# A Pilot Study of the Clinical Effectiveness of a Single Intra-Articular Injection of Stanozolol in Canines with Knee Degenerative Joint Disease and Its Correlation with Serum Interleukin-1β Levels

**DOI:** 10.3390/ani14091351

**Published:** 2024-04-30

**Authors:** L. Miguel Carreira, João Cota, Joao Alves, Filipa Inácio, Graça Alexandre-Pires, Pedro Azevedo

**Affiliations:** 1Anjos of Assis Veterinary Medicine Centre-CMVAA, Rua D.ª Francisca da Azambuja Nº9-9A, 2830-077 Barreiro, Portugalpedro.almeida.azevedo@gmail.com (P.A.); 2Faculty of Veterinary Medicine, University of Lisbon (FMV/ULisboa), Av. da Universidade Técnica, 1300-477 Lisbon, Portugal; finacio@fmv.ulisboa.com (F.I.); gpires@fmv.ulisboa.pt (G.A.-P.); 3Interdisciplinary Centre for Research in Animal Health (CIISA), University of Lisbon (FMV/ULisboa), Av. da Universidade Técnica, 1300-477 Lisbon, Portugal; 4Associate Laboratory for Animal and Veterinary Sciences (AL4AnimalS), 1300-477 Lisbon, Portugal; 5Faculty of American Laser Study Club—ALSC, Altamonte Springs, FL 32714, USA; 6Divisão de Medicina Veterinária, Guarda Nacional Republicana (GNR), Rua Presidente Arriaga, 9, 1200-771 Lisbon, Portugal; alves.jca@gnr.pt

**Keywords:** dog, degenerative joint disease, pain, anabolic, stanozolol, treatment

## Abstract

**Simple Summary:**

This study assessed the clinical efficacy of a single intra-articular stanozolol injection in canine knees with degenerative joint disease (DJD) and its correlation with serum IL-1β levels. Thirty dogs were divided into a control group (CG, n = 10) and a study group (SG, n = 20) with DJD. Pain levels were assessed using the Brown query, and radiographs were taken at T0 and T3. IL-1β levels were quantified via ELISA. Apart from 2 patients, all showed reduced pain intensity, with 15 showing improvement at T1 and 3 at T2. A positive correlation (r = 0.84; *p* < 0.01) was found between pain level and IL-1β in 15 patients. No systemic effects were observed. Most patients (18/20) experienced reduced pain, possibly linked to decreased catabolic IL synthesis and increased TGF-1β levels. This pilot study suggests stanozolol’s potential in managing DJD in dogs. Further research is warranted to validate these findings and understand stanozolol’s mechanism in DJD treatment.

**Abstract:**

Stanozolol shows promise as an anabolic and anti-catabolic agent for treating degenerative joint disease (DJD). This study assessed the clinical efficacy of a single intra-articular stanozolol injection in canine knees with DJD and its correlation with serum IL-1β levels. Thirty dogs (n = 30) were divided into a control group (CG, n = 10) and a study group (SG, n = 20) with DJD. Pain levels were assessed using the Brown query, and radiographs were taken at T0 and T3. IL-1β levels were quantified via ELISA. Apart from 2 patients, all showed reduced pain intensity, with 15 patients showing improvement at T1 and 3 patients at T2. A positive correlation (r = 0.84; *p* < 0.01) was found between pain level and IL-1β in 15 patients. No systemic effects were observed. Most patients (18/20) experienced reduced pain. This pilot study suggests stanozolol’s potential in managing DJD in dogs. Further research is warranted to validate these findings and understand stanozolol’s mechanism in DJD treatment.

## 1. Introduction

Degenerative joint disease (DJD), also known as osteoarthritis, is a common and debilitating condition characterized by progressive deterioration of articular cartilage, subchondral bone changes, and synovial inflammation. DJD primarily affects weight-bearing joints such as the hips, knees, and elbows, leading to pain, stiffness, and decreased mobility [1,2,3,4,5]. Brown’s original survey, the Canine Brief Pain Inventory (CBPI), developed in 2006 at the University of Pennsylvania, School of Veterinary Medicine, stands as a pivotal tool in assessing pain in dogs [6,7]. It provides a comprehensive evaluation of pain level and its impact on a dog’s daily functioning, offering a valuable means of communication between veterinarians and pet tutors. The CBPI comprises two sections: the pain severity score and the pain interference score. By capturing both the subjective experience of pain and its practical implications on the dog’s quality of life, the CBPI enables veterinarians to tailor treatment plans accordingly and monitor the effectiveness of interventions over time. The CBPI has been widely adopted in both clinical practice and research settings, contributing to advancements in the understanding and management of pain in dogs [6,7]. Several factors contribute to the development and progression of DJD, including genetic predisposition, abnormal joint conformation, obesity, trauma, and repetitive joint stress. The pathophysiology of the disease is complex and involves a intrincate interplay of biochemical and biomechanical processes, including increased production of proinflammatory cytokines, matrix metalloproteinases, and reactive oxygen species, which disrupt the delicate balance between cartilage synthesis and degradation [4,5,8]. Management of DJD in dogs typically involves multimodal approaches aimed at alleviating pain, improving joint function, and slowing disease progression, including pharmacological interventions, weight management, physical therapy, and in some cases, surgical options such as joint replacement. Early detection and intervention are crucial for optimizing outcomes and enhancing the quality of life for affected dogs [1,8,9,10]. Within the synovial membrane, both type A and type B synoviocytes, alongside dendritic cells, are discernible. Type A synoviocytes, derived from the bone marrow, constitute a part of the synovial mononuclear phagocytic system (SMNF) and are responsible for synthesizing Interleukin-1 (IL-1), prostaglandin E (PGE), and hyaluronic acid [11,12]. During inflammatory states, these cells additionally synthesize pro-inflammatory cytokines, thereby influencing chondrocyte metabolism [13]. Subsequently, chondrocytes become reactive, initiating the production of inflammatory mediators such as nitric oxide (NO), PGE2, and various cytokines including interleukin-1β (IL-1β) [14,15]. IL-1β stands as a pivotal cytokine in the pathogenesis of DJD, exerting a central role in mediating the inflammatory response and tissue degradation within affected joints. IL-1β stimulates synovial cells to secrete matrix metalloproteinases (MMPs) and other catabolic enzymes, leading to the breakdown of articular cartilage and underlying bone remodeling [16,17]. Moreover, it stimulates other cytokines (IL-6, IL-8), leukocyte inhibitory factor (LIF), and PGE2, perpetuating the inflammatory cascade and contributing to pain sensitization [4,15,16]. Initially synthesized as an inactive precursor, IL-1β is converted into its active form by the enzyme caspase 1 or IL-1β converting enzyme (ICE), whose activity is heightened in DJD [15].Elevated levels of IL-1β have been consistently observed in the synovial fluid and cartilage of dogs with DJD, correlating with disease severity and progression [18,19,20]. The anti-inflammatory and analgesic properties inherent in non-steroidal anti-inflammatory drugs (NSAIDs), corticosteroids, and opioids, whether administered orally or intra-articularly, render them primary choices for treating degenerative joint disease (DJD) in dogs [21]. However, the emergence of adverse effects across multiple organic systems is a stark reality, particularly considering the frequent repetition of treatment protocols in most cases [22,23,24]. Despite varying degrees of symptom modulation afforded by NSAIDs, corticosteroids, and opioids, these pharmacological agents fail to impede changes in cartilage and joint structure [21]. The nonsteroidal anti-inflammatory drugs (NSAIDs) are cornerstone therapeutics in managing DJD-associated pain and inflammation [25]. NSAIDs exert their effects by inhibiting cyclooxygenase (COX) enzymes, thereby reducing the synthesis of prostaglandins, which are key mediators of inflammation and pain. While NSAIDs provide symptomatic relief, they also carry potential risks, including gastrointestinal, renal, and hepatic adverse effects, necessitating cautious use and regular monitoring. In conjunction with NSAID therapy, multimodal treatment protocols are often employed to address the multifaceted nature of DJD [24,26,27]. These protocols may include physical rehabilitation, weight management, nutraceutical supplementation (such as glucosamine and chondroitin sulfate), disease-modifying osteoarthritis drugs (DMOADs), such as polysulfated glycosaminoglycans (PSGAGs), and adjunctive pain management modalities like acupuncture and laser therapy. By combining various therapeutic modalities, veterinarians can tailor treatment regimens to individual patients, aiming not only to alleviate pain and inflammation but also to preserve joint function and enhance the overall quality of life for dogs affected by DJD [25,26,27,28]. Ongoing investigations delve into novel molecules possessing anabolic and anti-catabolic attributes within cartilage, aimed at either mitigating or ameliorating DJD. Stanozolol a synthetic derivative of testosterone, emerges as one such molecule, possessing both anabolic and anti-catabolic properties with potential therapeutic benefits in the treatment of DJD in dogs. Its anabolic effects promote tissue repair and regeneration, which could aid in mitigating cartilage degradation and promoting joint health. Additionally, stanozolol possesses anti-catabolic properties, potentially inhibiting the activity of matrix metalloproteinases (MMPs) and other catabolic enzymes involved in cartilage breakdown. Although research on the use of stanozolol specifically for DJD in dogs is limited, with only few in vitro and in vivo animal studies published, all concluded for its potential as a promising therapeutic agent for symptomatic relief in DJD, presenting efficacy in improving joint function and reducing pain with purported regenerative effects within the joint [29,30,31,32,33,34,35,36,37,38]. This study aims to evaluate the clinical efficacy of a single intra-articular stanozolol infiltration in treating canine knees afflicted with DJD, while also investigating it association with fluctuations in serum IL-1β levels in dogs with DJD both before and after treatment with stanozolol.

## 2. Materials and Methods

The study utilized a convenience sampling of 30 (N = 30) inpatient dogs evaluated at Anjos of Assis Veterinary Medicine Centre (CMVAA), encompassing both genders, divided into two distinct groups: the study group (SG), comprising 20 dogs diagnosed with degenerative joint disease (DJD), and the control group (CG), consisting of 10 healthy individuals. Convenience sampling allowed us to efficiently recruit dogs with degenerative joint disease (DJD) who met the inclusion criteria and were suitable for elective treatment, without imposing additional demands on resources or tutors within the hospital setting. The study used a quasi-experimental design because the allocation of dogs to the treatment group and the control group was not based on random assignment but rather on practical clinical considerations and availability. Quasi-experimental designs can provide valuable insights into the effects of interventions in real-world settings. Approval for the study was obtained from the Animal Ethics and Welfare Council (CEBEA) of FMV-ULisboa under the reference number 040/2018. Participation of animals commenced only subsequent to the owners’ signing of informed consent forms. To minimize potential bias associated with convenience sampling, clear inclusion criteria were established to ensure that only dogs diagnosed with DJD in one knee joint confirmed through digital radiographs encompassing mediolateral and craniocaudal views, and undergoing therapy with chondroprotectors and gabapentin were included in the study. This helped to maintain homogeneity within the sample and reduce the risk of including dogs with other unrelated conditions. Additionally, patients were required to be receptive to monitoring at multiple time points as pre-established in the protocol, with tutors capable of providing information regarding their pets’ evolving mobility. The CG consisted of 10 healthy animals devoid of recent or chronic diseases, with normal hemogram readings and basic liver and kidney biochemistries. The study design encompassed four time-points: T0 (initiation of the protocol), T1 (4th day post-injection), T2 (8th day post-injection), and T3 (23rd day post-injection), with various tasks undertaken at each time-point (Table 1 for details).

The monitoring of patients’ clinical progression relied on data derived from questionnaires completed by owners and forms documented by researchers (Appendix A is presented in the Appendix A). Owners were presented with a validated questionnaire developed by Brown (2006) [6] to assess chronic pain and quality of life in dogs.

The outcomes of each questionnaire were evaluated to determine the correlation between owners’ perceptions and the patients’ quality of life, as well as the patients’ pain levels and its respective impact on function. Several parameters from the questionnaire were integrated into the form used by clinicians, including: pain level upon joint manipulation, degree of lameness, and rectal and transcutaneous knee joint temperatures.

All joints underwent radiographic examination before infiltration at T0 and at T3 (after 23 days) to assess morphological changes following the protocol established by Innes (2010) [23] for evaluating DJD in dogs’ knees, which considers the following parameters that are quantified: global disease status (0–3), joint effusion (0–2), osteophytosis (0–3), intra-articular mineralization (0–2), and subchondral sclerosis (0–1). Evaluation of images was conducted by the same two researchers to mitigate bias, and each examination included craniocaudal and mediolateral views. 

Transcutaneous joint temperature was measured using an infrared technology thermometer (Thermofocus^®^—Tecnimed, Sintra, Portugal) after shaving the fur around the joint of the knee. Measurements were taken laterally to the patellar tendon, duplicated for accuracy, and averaged. Rectal temperature was also recorded twice for each patient to obtain the mean temperature.

Peripheral blood samples were collected from each participant to quantify IL-1β levels using an ELISA method, recognized as a key mediator of catabolism and joint inflammation in DJD (Buckwalter et al., 2005) [39]. In the CG, this procedure was conducted solely at T0, whereas in the GDJD group, it was performed at both T0 and T2. Venous punctures were executed at the cephalic or saphenous veins using 23G needles coupled with 2 mL syringes and after the topical application of bupivacaine gel on the skin. Subsequently, blood samples were centrifuged, and the serum separated and stored at –20 °C until analysis. For treatment, an injectable aqueous solution containing 50 mg/mL of Stanozolol, with specified excipients, was utilized. Each treated joint received a single dose at T0 of 0.3 mL/kg. Prior to arthrocentesis, the region was prepared with bupivacaine gel followed by subcutaneous infiltration of lidocaine using a 25G needle. Arthrocentesis was performed with a 23G needle without complications, and passive mobilization of each joint ensued to facilitate drug dispersion. None of the patients required anesthesia or sedation during the procedure. IL-1β quantification was performed using an ELISA system from Sigma-Aldrich^®^Darmstadt, Germany, validated for measuring canine IL-1β in various biological samples (serum, plasma and cells supernatantes). All manufacturer’s instructions were strictly followed, including simultaneous duplicate analysis of all samples and absorbance measurement at 450 nm using a Tecan brand spectrophotometer (Spectra Classic Plate Reader).

Statistical analysis was conducted using SPSS© Statistics 19 from IBM© (Armonk, NY, USA) and MedCalc^®^ 15 from BVBA (Antwerp, Belgium). Descriptive statistics; testing for normality of variables related to the ages of the individuals under study, as well as their weights and concentrations of IL-1β; evaluation of inter-rater agreement (between clinicians, or between clinicians and tutors); and calculation of the differences of significance differences between variables from different groups or different time points were all performed. Statistical tests, including the Shapiro Wilk test, paired *t*-test, Wilcoxon test, Friedman test, Welch equation, and the Pearson correlation coefficient were conducted. Concordance between different opinions was calculated using the Lin method and Cronbach’s Alpha. All statistical analyses assumed a significance level of 95% with a *p*-value < 0.05.

## 3. Results

The study sample comprised both genders, with 65% females and 35% males. In the control group (CG), 61.5% were females and 31.8% were males, with a mean age of 3.9 ± 0.7 years (range: 1 to 9 years) and an average body weight of 24.3 ± 3 kg (range: 11 to 45 kg). Normality was observed for age (*p* = 0.28) and body weight (*p* = 0.31) based on the Shapiro-Wilk test. In the study group (SG), 71.5% were females and 29% were males, with a mean age of 8.7 ± 0.7 years (range: 6.4 to 12.1 years) and an average body weight of 33.3 ± 4.9 kg (range: 20 to 52.7 kg).

Degenerative joint disease (DJD) was identified in the left knee of 9 individuals and in the right knee of 11 individuals. Normality was confirmed for age (*p* = 0.27) and body weight (*p* = 0.61) using the Shapiro-Wilk test. Statistically significant differences were observed between the CG and the SG regarding age (*p* = 0.0003) but not for body weight (*p* = 0.15) according to the Welch equation.

Three SG patients and two CG animals had serum IL-1β concentrations below the limit of detection (>10 pg/mL) at all time points. IL-1β serum concentrations followed a normal distribution in the CG (*p* = 0.33), with a mean of 54.54 pg/mL (maximum: 98.18 pg/mL; minimum: 10.91 pg/mL, measured only at T0).

In the SG, IL-1β serum concentrations also followed a normal distribution at T0 (*p* = 0.56), with a mean of 108.18 pg/mL (maximum: 192.73 pg/mL; minimum: 36.36 pg/mL) and at T2 (mean: 79.77 pg/mL; *p* = 0.52; maximum: 140.91 pg/mL; minimum: 36.36 pg/mL). No statistically significant differences were noted for IL-1β serum concentrations between T0 and T2 based on paired-sample *t*-test (*p* = 0.56). Additionally, using the Welch equation, no statistically significant differences were observed between the CG and the SG (*p* = 0.20) (Figure 1).

According to the owners’ assessment, with the exception of 3 patients, all others exhibited a reduction in pain intensity at some point during the protocol, and in 13 animals, this reduction persisted until the end of the protocol (23rd day). The impact of pain on function, as evaluated by the owners, also showed favorable variation in all individuals, except for the initial 3 patients where no changes were noted (Figure 2). Regarding the parameter of quality of life, as reported by the owners, all animals demonstrated a subtle improvement except for 3 patients.

During clinical assessment, the pain level resulting from joint manipulation decreased in all SG patients, except for the same 3 patients, throughout the protocol. The pain level during knee manipulation followed a normal distribution at all time points: T0 (*p* = 0.09), T1 (*p* = 0.68), T2 (*p* = 0.47), and T3 (*p* = 0.47). In the SG, 15 animals showed improvement in pain level at T1 (first re-evaluation), 3 animals only at T2, and 2 animals showed no improvement at T3 (Figure 3).

Variations in pain levels during patient manipulation exhibited statistically significant differences between T0 and T2 (*p* = 0.02) and between T0 and T3 (*p* = 0.03) according to paired-sample *t*-tests. Pearson’s correlation coefficient between IL-1β serum concentration and pain levels during joint manipulation was positive (r = 0.84) and statistically significant (*p* < 0.01) in 17 patients (SG animals with IL-1β serum concentrations above the 10 pg/mL detection limit). Assessment of pain’s influence on mobility functions showed an improvement in life quality in all SG individuals, except for 2 patients (Figure 3). Concordance correlation evaluation between parameters assessed by clinicians and owners at each time point was conducted using the Lin method. A moderate concordance (Pc > 0.90) was observed between owners’ and clinicians’ opinions regarding pain levels and pain during joint manipulation, but only at T0. The agreement between owners’ and clinicians’ opinions regarding pain’s influence on patient motion and life quality was weak at all time points (Table 2).

At the onset of the protocol, all SG patients exhibited lameness. Throughout the study’s duration, 15 patients demonstrated very significant improvements, while 3 patients showed moderate improvement. 

Radiographic interpretations were conducted blindly by three evaluators to mitigate bias. The Cronbach’s alpha test revealed a high level of agreement between film evaluations at T0 (α = 0.94) and T3 (α = 0.92). The Wilcoxon nonparametric test for paired samples was employed to analyze the set radiographic parameters, and among all evaluated items, only joint effusion exhibited statistically significant variation (*p* < 0.00) between T0 and T3. 

The mean transcutaneous joint temperature was 37.6 °C at T0 and 37.4 °C at T1, both following a normal distribution (*p* = 0.60 for T0 and *p* = 0.33 for T1). The mean rectal temperature was 39.4 °C at T0 and 38.7 °C at T1, also exhibiting normal distribution (*p* = 0.33 for T0 and *p* = 0.67 for T1). Using the *t*-test for paired samples, no statistically significant differences were observed between T0 and T1 for transcutaneous joint temperature (*p* = 0.75) and rectal temperature (*p* = 0.45).

## 4. Discussion

In the study, a convenience sampling of 30 dogs was chosen, which can be justified when the study population is difficult to access or limited in availability, as was the case in the study conducted within a hospital environment. Convenience sampling enabled the efficient recruitment of dogs with degenerative joint disease (DJD) who met the inclusion criteria and were suitable for elective treatment, without imposing additional demands on resources or staff within the hospital setting. However, to minimize potential biases associated with convenience sampling, several steps were taken: (1) inclusion criteria: Clear inclusion criteria were established to ensure that only dogs diagnosed with DJD were included in the study. This helped to maintain homogeneity within the sample and reduce the risk of including dogs with other unrelated conditions, (2) standardized assessment: All dogs underwent standardized assessments and diagnostic procedures to confirm the presence of DJD and assess eligibility for treatment. This helped to ensure consistency in the evaluation process and minimize variability in patient selection, (3) transparency: The rationale for choosing convenience sampling was clearly stated in the study protocol and manuscript. Transparency about the sampling method helps readers understand the limitations of the study and interpret the results accordingly, and (4) statistical analysis: Statistical techniques, such as sensitivity analysis or propensity score matching, were employed to adjust for potential biases inherent in convenience sampling. These methods help to mitigate the impact of confounding variables and strengthen the validity of the study findings. Moreover, emphasis was placed on a balanced interpretation of the findings in regards to result interpretation, thereby enhancing the credibility and reliability of the study. Overall, while convenience sampling may introduce certain biases, careful consideration of study design, rigorous methodology, and transparent reporting can help to minimize these biases and strengthen the validity of the research findings. Regarding to age parameter, the observed difference in mean age between the SG and CG animals aligns with the typical age prevalence of degenerative joint disease (DJD) in dogs, which commonly affects individuals over 4 years old [23,40]. Four specific time points were selected in the study for quantifying IL-1β concentrations following intra-articular stanozolol injection. The selection was based on several scientific considerations. Firstly, time point 0 (time of injection) serves as the baseline measurement, allowing for comparison with subsequent time points to assess changes in IL-1β levels over time. Time points 1 (4 days after injection), 2 (8 days after injection), and 3 (21 days after injection) were chosen to capture both short-term and medium/long-term effects of stanozolol treatment on IL-1β concentrations within the joint. At time point 1 (4 days after injection), it is anticipated that early molecular and cellular responses to stanozolol administration may begin to manifest, potentially leading to alterations in IL-1β levels as part of the inflammatory cascade. Selection of day 8 post-injection (T2) to quantify IL-1β levels was justified by the fact that at this time point, variations in IL-1β may reflect the early response to stanozolol treatment and provide insights into its initial effects on inflammatory processes within the joint. At this time point (T2), it could be particularly relevant for assessing short-term changes in IL-1β expression and evaluating the early efficacy of stanozolol in modulating inflammation. Additionally, by time point 3 (21 days after injection), it is expected that the full extent of stanozolol’s impact on IL-1β levels, including any potential medium/long-term effects or resolution of inflammation, can be assessed. This selected time frame it aligns with previous research methodologies examining the pharmacokinetics and pharmacodynamics of intra-articular therapies in veterinary medicine [41,42,43,44,45]. 

While stanozolol has been utilized in various animal species, including horses, sheep, and dogs, a specific therapeutic dosage for intra-articular DJD treatment has yet to be established [36,37,46,47]. In prior work by Carli [38], a dose of 1.5 mg per joint was administered fortnightly in dogs, without considering individual body weight. However, in our study, stanozolol was administered as a single dose at 0.3 mg/kg, adjusting for each patient’s body weight. This dosing regimen was derived from the study by Adamama-Moraitou et al. [47] regarding stanozolol use in dogs. Notably, none of the 20 individuals receiving intra-articular infiltration exhibited systemic anabolic or androgenic effects, consistent with prior findings on both systemic and intra-articular stanozolol administration [36,37,38,46,47,48]. Evaluation of pain levels by tutors throughout the protocol consistently indicated improvement, with pain upon joint manipulation deemed the most reliable parameter, assessed exclusively by trained technicians. The majority of patients (18 out of 20) exhibited a decrease in pain level in the infiltrated joint, without any experiencing worsening compared to their initial clinical condition. The studies conducted by Carli [38] and Spadari et al. [48] have documented instances where patients initially experienced a worsening of symptoms following intra-articular stanozolol infiltration, with clinical improvements observed only after this transient exacerbation. This phenomenon has been attributed to a temporary increase in the inflammatory component of degenerative joint disease (DJD) triggered by stanozolol’s high molecular weight. Interestingly, such symptom aggravation was not observed at any point in our present study. The statistically significant variation in pain level during joint manipulation between T0T2 and T0T3 suggests the presence of an analgesic effect of stanozolol in DJD treatment, consistent with findings from prior research [36,37,38]. This analgesic property may stem from stanozolol’s effect on rheumatoid arthritis (RA), coupled with its anti-catabolic action at glucocorticoid receptors. These mechanisms are believed to reduce the synthesis of catabolic interleukins (ILs) and promote the normalization of synovial fluid properties [48]. Moreover, stanozolol has been associated with increased transforming growth factor-1β (TGF-1β) concentration, which has been linked to reduced joint pain in humans [37], as well as the inhibition of nitric oxide (NO) production by chondrocytes, which contributes to the perpetuation of the DJD catabolic cycle [15,23,36]. Our study focused on evaluating the effects of a single intra-articular stanozolol treatment in dogs with DJD and its correlation with serum interleukin-1β (IL-1β) concentration variations during treatment. The results demonstrate the efficacy of stanozolol in managing DJD in dogs, with improvements in pain levels observed in 15 patients at T1 (4 days after infiltration) and in 3 animals at T2 (8 days after infiltration). The observed variation in treatment response among patients with degenerative joint disease (DJD) can be attributed to several factors. Notably, patients in the acute phase of DJD tend to benefit most from stanozolol therapy for pain relief, as the inflammatory component is likely the predominant contributor to pain during this phase [23]. In contrast, patients in a quiescent phase of DJD may experience more discrete or no improvement, as other factors such as neuropathic pain may also be present. Interestingly, patients who initially exhibited a pain level score on joint manipulation equal to or higher than 6 showed a minimum and maximum improvement of 2 and 4 points, respectively, on the pain scale. Conversely, patients with lower initial pain levels experienced smaller improvements, with some showing no change. This suggests that the severity of pain at the outset may influence the degree of improvement with stanozolol therapy. Furthermore, patients with the greatest reduction in pain levels also exhibited better radiographical joint conditions compared to others, despite initially higher pain levels. This correlation between pain reduction and improved joint condition supports the hypothesis that stanozolol’s therapeutic effect includes a significant anti-inflammatory component. Radiographically, favorable evolution of the joint effusion parameter between T0 and T3 further supports the anti-inflammatory properties of stanozolol. However, it remains challenging to determine whether pain level or morphological joint changes are the primary limiting factors in functional changes observed in patients with DJD. Overall, the observed variations underscore the complexity of DJD management and highlight the need for tailored treatment approaches based on individual patient characteristics and disease phase. Further research is warranted to elucidate the underlying mechanisms driving treatment response and optimize therapeutic strategies for DJD in dogs. The study’s findings suggest that the assessment of functional changes in patients with degenerative joint disease (DJD) may be influenced by various factors beyond pain level alone. While a decrease in pain level was observed throughout the protocol, improvements in function did not always parallel the magnitude of pain reduction. This raises the hypothesis that pain level may be initially overvalued in the assessment of functional changes, highlighting the complexity of evaluating DJD patients’ mobility and quality of life. Changes in lameness patterns over the study period were not statistically significant and did not consistently correlate with decreases in pain level. This lack of linear correlation may be attributed to morphological joint changes, muscle atrophy, and contracture observed in some patients, which can affect mobility independently of pain level reduction. Additionally, agreement between clinicians and tutors regarding the assessment of pain level interference in animal function was low, indicating differences in awareness between the two groups regarding the role of morphological changes in functional limitations. This underscores the importance of considering both subjective assessments of pain and objective measures of joint health when evaluating DJD patients’ functional status. The study demonstrated that the administration of stanozolol injections has yielded promising improvements in both pain management and quality of life outcomes in dogs with degenerative joint disease (DJD). Clinical studies in dog and other species have reported significant reductions in pain scores and enhanced mobility following stanozolol treatment [2,36,49,50,51,52]. Additionally, tutors often observe improvements in their pets’ activity levels and overall well-being, indicating a positive impact on quality of life. While statistical analyses confirm these improvements, questions may arise regarding their clinical significance. To establish whether the observed improvements are clinically meaningful or merely a statistical artifact, it is essential to consider factors such as the magnitude of improvement, duration of effects, and impact on daily functioning and long-term prognosis. Further research, including randomized controlled trials and longitudinal studies, is necessary to validate the clinical significance of stanozolol therapy in dogs with DJD and other musculoskeletal disorders. Thermographic studies play a pivotal role in veterinary medicine, particularly in evaluating changes in joint temperature. This non-invasive technique utilizes infrared technology to detect subtle alterations in surface temperature associated with various joint conditions. By visualizing thermal patterns, veterinarians can assess inflammation, injury, and dysfunction within joints, aiding in early detection and accurate diagnosis. Thermography offers numerous advantages, including real-time monitoring, objective assessment of treatment response, and reduced stress for animals. Veterinary research has demonstrated the efficacy of thermography in diagnosing joint inflammation in dogs, and evaluating osteoarthritis in small animals. These studies underscore the value of thermography as a valuable adjunctive tool in veterinary diagnostics, enhancing our ability to provide optimal care for our animal patients [53,54,55,56,57]. In our study, transcutaneous and rectal temperature measurements were obtained using an infrared technology thermometer and aimed to detect joint temperature variations associated with inflammation. The analysis did not reveal significant decreases between T0 and T1 (4 days post injection), making it challenging to directly associate stanozolol with an effect on joint temperature. However, standardizing body temperature measurements across patients proved challenging due to differences in dimensions and metabolism. Despite efforts to acclimatize patients to a consistent temperature environment, variations in body temperature persisted, complicating the interpretation of temperature data in relation to joint inflammation. It’s suggested that the use of a thermographic camera insteat a infrared thermometer could provide a more nuanced understanding of the relationship between local temperatura and the stnozolol, as demonstrated in previous studies with other steroids [58,59,60]. 

The study employed a specific validated ELISA system for serum and plasma canine IL-1β, including a control group (CG) to compare IL-1β levels with the stanozolol group (SG), given the lack of reference values for IL-1β in the literature. The decision was made to test for serum IL-1β instead of synovial fluid because the test indicated its performance with serum and plasma, but not with synovial fluid, due to specific restrictions of the kit or assay validation. Additionally, it is more difficult to collect a sufficient volume of synovial fluid than to obtain a serum sample for use in the ELISA kit used in the study. Furthermore, serum IL-1β levels can provide systemic information about the inflammatory response, reflecting overall joint health and potentially capturing systemic effects of treatment. Moreover, serum IL-1β levels may correlate with disease severity and treatment response, providing valuable insights into the efficacy of therapeutic interventions, namely to the use of stanozolol. While synovial fluid analysis offers direct information about the local inflammatory milieu within the joint, serum IL-1β assessment offers a broader perspective that may be more representative of the overall inflammatory status of the patient. Moreover, it is important that IL-1β concentrations were below the detection limit in three SG patients. While this contrasts with findings from another study by Prachar, Kaup, and Neumann [61], where IL-1β concentrations were detected in healthy dogs; it’s important to note the different methodologies employed between the two studies. Moreover, it important to notice that while serum IL-1β is implicated in the inflammatory cascade associated with DJD, its levels may vary due to factors such as disease severity, individual variability in inflammatory response, and assay sensitivity limitations. Consequently, relying solely on serum IL-1β for DJD diagnosis may lead to false-negative results. A comprehensive diagnostic approach, integrating clinical evaluation, imaging modalities (e.g., radiography, MRI), and multiple biomarkers, is warranted for accurate DJD diagnosis and monitoring. Further research exploring the utility of serum IL-1β alongside other diagnostic markers is necessary to enhance diagnostic accuracy in dog DJD [62,63,64,65,66,67,68]. Although differences in IL-1β quantifications between SG and CG at T0 were not statistically significant, the mean serum IL-1β concentration was higher in SG compared to CG. SG individuals showed a decrease in serum IL-1β at T2, suggesting a potential local anti-inflammatory effect of stanozolol. The positive correlation observed between serum IL-1β concentrations and pain level upon joint manipulation further supports this hypothesis, indicating the presence of a significant inflammatory component in pain. The study suggests that while some individuals may experience a reduction in the inflammatory component, others may have pain originating from alternative pathways, such as the neuropathic pathway. The variability of the DJD inflammatory component is highlighted, with the disease evolving mainly at the expense of agents other than IL-1β in the final stages, as described in the literature [14,15,67].

Considering the statistical analysis, the use of multiple *t*-tests without adjustment was based on our research objectives, design, and practical considerations. Various outcome measures were examined across different time points and conditions, and *t*-tests were conducted to test specific hypotheses. Given the exploratory nature of the study, adjusting for multiple comparisons could increase the risk of type II statistical errors. By using non-parametric tests alongside parametric tests, robustness in the results was ensured. While acknowledging potential type I errors, the interpretation of results was done cautiously, emphasizing effect sizes and overall patterns rather than relying solely on *p*-values. The inclusion of non-parametric tests adds validation and complements parametric findings [69,70,71,72,73,74]. Overall, the study highlights the multifactorial nature of DJD assessment and the importance of considering various factors, including pain level, functional changes, and joint morphology, when evaluating treatment outcomes and designing therapeutic interventions for DJD patients. Further research is needed to better understand the complex interplay between these factors and optimize treatment strategies for DJD in dogs.

## 5. Conclusions

The pilot study aimed to assess the effects of a single intra-articular injection of stanozolol in dogs with DJD and its association with serum IL-1β variation during treatment. While the study did not extend to evaluating long-term changes in serum IL-1β levels and the effects of stanozolol on joint morphology over an extended follow-up period posttreatment, the findings suggest that stanozolol may be beneficial in managing DJD in dogs. The observed improvements in pain levels upon joint manipulation and the reduction in serum IL-1β concentrations support the potential anti-inflammatory effects of stanozolol in treating DJD. Additionally, the lack of systemic anabolic or androgenic effects observed in the study aligns with previous research on stanozolol’s safety profile in dogs. Further research with a larger sample size and longer follow-up periods could provide additional insights into the efficacy and safety of stanozolol in managing DJD in dogs. Additionally, exploring its effects on joint morphology and inflammation over time could contribute to a more comprehensive understanding of its therapeutic potential in treating this condition.

## Figures and Tables

**Figure 1 animals-14-01351-f001:**
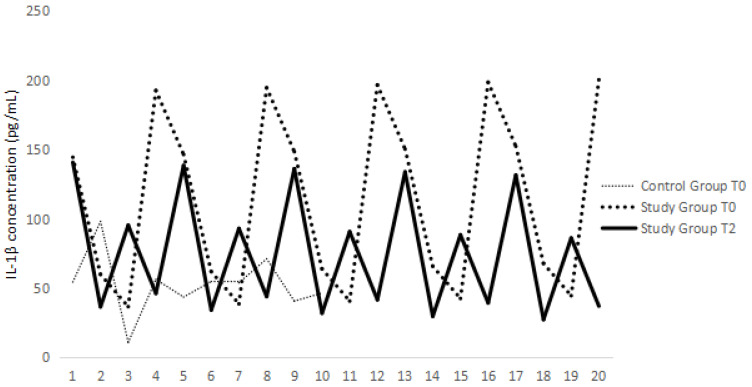
IL-1β (pg/mL) variations on the control group and the Study Group at T0 and T2.

**Figure 2 animals-14-01351-f002:**
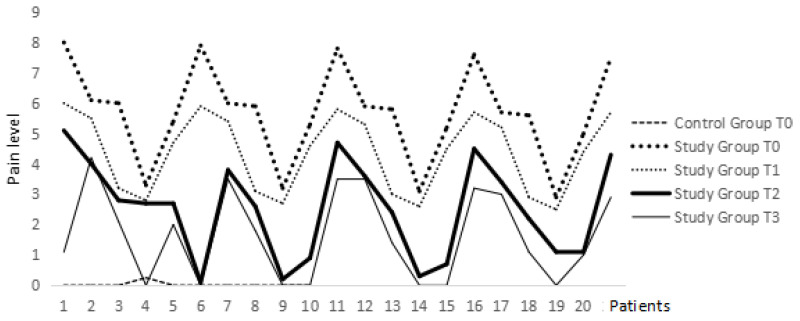
Assessment of pain level interference on patient function according to tutors.

**Figure 3 animals-14-01351-f003:**
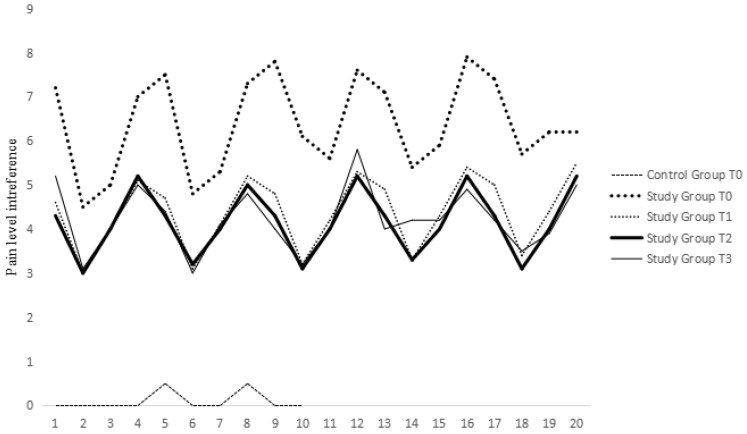
Clinical pain assessment during joint manipulation of the patients.

**Table 1 animals-14-01351-t001:** Checklist of the tasks developed during the study process over the four time points considered (T0–T3).

Time Points	Tasks
T0 (day of injection)	✓ Assessment of pain level and quality of life in dogs by tutors ✓ Assessment of pain level by clinicians ✓ Rectal and transcutaneous temperature ✓ Craniocaudal and mediolateral radiographs to the knee ✓ Blood sample for IL-1β ✓ Stanozolol intra-articular injection
T1 (4th day after injection)	✓ Assessment of pain level and quality of life in dogs by tutors ✓ Assessment of pain level by clinicians ✓ Rectal and transcutaneous temperature
T2 (8th day after injection)	✓ Assessment of pain level and quality of life in dogs by tutors ✓ Assessment of pain level by clinicians ✓ Blood sample for IL-1β
T3 (23rd day after injection)	✓ Assessment of pain level and quality of life in dogs by tutors ✓ Assessment of pain level by clinicians ✓ Craniocaudal and mediolateral radiographs to the knee

**Table 2 animals-14-01351-t002:** Concordance correlation between the parameters evaluated by clinicians and the tutors at each time point carried out using the Lin method.

Concordance Correlation	Lin Method	Study Time Points			
		T0	T1	T2	T3
Tutors and Clinicians opinion regarding pain level and pain on joint manipulation	Pc	0.914 *	0.634	0.831	0.768
Tutors and Clinicians opinion regarding pain influence in patient motion	Pc	0.046	0.338	0.391	0.373
Tutors and Clinicians opinion regarding patient life quality	Pc	0.176	0.432	0.432	0.774

*** Statistically significant.

## Data Availability

The raw data supporting the conclusions of this article will be made available by the authors on request.

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
