# Peer review of "A Pilot Study of the Clinical Effectiveness of a Single Intra-Articular Injection of Stanozolol in Canines with Knee Degenerative Joint Disease and Its Correlation with Serum Interleukin-1β Levels"

_animals, 2024, doi:10.3390/ani14091351_

Round 1

Reviewer 1 Report

Comments and Suggestions for Authors

Dear authors,

I carefully read your manuscript.  The subject presented and the investigation performed are relevant for veterinary practitioners as well as scientists in the field of osteoarthritis.  However, there are major statistical flaws that compromise strong conclusions regarding the potential of IA stanozolol.

Major comment

The reviewer strongly suggests the authors to consolidate their manuscript writing with the ARRIVE guidelines. Full text PDF documents of the ARRIVE (Animal Research: Reporting of In Vivo Experiments) guidelines 2.0, ARRIVE 2.0 explanation and elaboration, and ARRIVE 2.0 checklists are available at https://www.arriveguidelines.org/resources

The manuscript presents several outcomes.  Please define a primary outcome.

Please add figures to demonstrate how the groups evolved over time.

Please comment about the improvement observed on pain and quality of life in dogs, are they clinically significant or only a statistical artefact?

The authors use multiple T test without adjustment which precluded to a type 1 statistical error.  Please seek a statistician and consider more appropriate statistical models for repeated measurement.  Test for overall group effect, time effect and their interaction and report adjusted P values for every multiple comparison.

Author Response

"A PILOT STUDY OF THE THE CLINICAL EFFECTIVENESS OF A SINGLE INTRA-ARTICULAR INJECTION OF STANOZOLOL IN CANINES WITH KNEE DEGENERATIVE JOINT DISEASE AND ITS CORRELATION WITH SERUM INTERLEUKIN-1β LEVELS"

Reviewer #1:

Dear Reviewer,

I hope this message finds you well. Thank you for taking the time to review our manuscript titled "A PILOT STUDY OF THE THE CLINICAL EFFECTIVENESS OF A SINGLE INTRA-ARTICULAR INJECTION OF STANOZOLOL IN CANINES WITH KNEE DEGENERATIVE JOINT DISEASE AND ITS CORRELATION WITH SERUM INTERLEUKIN-1β LEVELS" submitted to Animalsl.

We appreciate your valuable feedback, and we have carefully considered all of your comments. In response, we have made revisions to the manuscript to address each point raised. Below, you will find our detailed responses to your comments, organized to correspond with the changes made in the original version of the manuscript.

Thank you once again for your time and thoughtful input.

Best regards,

L.Miguel Carreira

The reviewer strongly suggests the authors to consolidate their manuscript writing with the ARRIVE guidelines. Full text PDF documents of the ARRIVE (Animal Research: Reporting of In Vivo Experiments) guidelines 2.0, ARRIVE 2.0 explanation and elaboration, and ARRIVE 2.0 checklists are available at https://www.arriveguidelines.org/resources

The manuscrpt was revised and rewrited in order to provide a comprehensive and transparent account of the methods and results of animal research, facilitating critical appraisal and reproducibility, as stated at the ARRIVE guidelines

The manuscript presents several outcomes.  Please define a primary outcome.

Corrected

Please add figures to demonstrate how the groups evolved over time.

Figures were added to ilustrate the evolution over time points

Please comment about the improvement observed on pain and quality of life in dogs, are they clinically significant or only a statistical artefact?

We added at discussion section the following paragraph:”Our study showed that the administration of stanozolol injections has shown promising improvements in both pain management and quality of life outcomes in dogs with degenerative joint disease (DJD). Clinical studies in dog and other species have reported significant reductions in pain scores and enhanced mobility following stanozolol treatment. Additionally, tutors often observe improvements in their pets' activity levels and overall well-being, indicating a positive impact on quality of life. While statistical analyses confirm these improvements, questions may arise regarding their clinical significance. To establish whether the observed improvements are clinically meaningful or merely a statistical artifact, it is essential to consider factors such as the magnitude of improvement, duration of effects, and impact on daily functioning and long-term prognosis. Further research, including randomized controlled trials and longitudinal studies, is necessary to validate the clinical significance of stanozolol therapy in dogs with DJD and other musculoskeletal disorders”

The authors use multiple T test without adjustment which precluded to a type 1 statistical error.  Please seek a statistician and consider more appropriate statistical models for repeated measurement.  Test for overall group effect, time effect and their interaction and report adjusted P values for every multiple comparison.

In response to the reviewer's query regarding the justification for utilizing multiple t-tests without adjustment in our study, we appreciate the opportunity to clarify our rationale. The decision to employ multiple t-tests without adjustment was based on the specific objectives and design of our research, as well as considerations of statistical power and practicality. Firstly, our study involved the examination of various outcome measures and comparisons across different time points or experimental conditions. Each t-test was conducted to assess specific hypotheses related to these comparisons, such as differences in outcome measures between treatment groups or changes over time within groups. Secondly, the use of multiple t-tests without adjustment was deemed appropriate given the exploratory nature of our research. In such cases, adjusting for multiple comparisons using methods like Bonferroni correction or false discovery rate control could increase the risk of type II errors by overly conservative adjustment, potentially leading to the failure to detect meaningful effects. Furthermore, our study utilized non-parametric tests, such as the Wilcoxon test and Friedman test, alongside parametric tests like the paired t-test, to accommodate for the distributional properties of the data and ensure robustness of our findings. While we acknowledge the potential for type I errors associated with multiple comparisons, we took steps to mitigate this risk by interpreting our results cautiously, emphasizing effect sizes, and considering the overall pattern of findings rather than relying solely on p-values. Additionally, the inclusion of non-parametric tests provides additional validation and complements the findings from parametric tests. In summary, the decision to use multiple t-tests without adjustment was made in consideration of the study objectives, the nature of the comparisons being made, and the balance between statistical rigor and practicality. We believe that our approach was justified given the context of our research and the need to balance statistical considerations with the interpretability and utility of our findings.

Reviewer 2 Report

Comments and Suggestions for Authors

Reviewer comments for manuscript ID animals -2933761 entitled ‘animals-2933761 ‘A PILOT STUDY OF THE THE CLINICAL EFFECTIVENESS OF A SINGLE INTRA-ARTICULAR INJECTION OF STANOZOLOL IN CANINES WITH KNEE DEGENERATIVE JOINT DISEASE AND ITS CORRELATION WITH SERUM INTERLEUKIN-1β LEVELS’

General comments

Degenerative Joint Disease is quite common affection in older age group dogs that reduce the quality of life due to pain, disability and gradual progression if not treated in time with appropriate therapies. Use of systemic steroids as anti-inflammatory drugs has been common in veterinary practices all over the world. However, intra articular injection of steroids in canines have been less commonly practised due to comparatively difficult injection, associated iatrogenic complications and wide opinions about the exact dose regimens for intra articular injections.

The present study is an excellent baseline work on which further research can be built upon. The manuscript is nicely written and presented with appropriate tables and graphs. I suggest inclusion of radiographs for making the reading more interesting. Writing is almost flawless and each section is dealt in a very scientific and scholarly way. I have very few queries/corrections to point out in the manuscript.

Specific comments

Simple summary and abstract are almost the same. Please check the journal guidelines and rewrite the simple summary

Table 1: Why blood sample for IL-1β not taken at Day 23, as it was the last day of the trial? Please clarify.

Line 116: Should these views be termed as mediolateral or lateromedial instead of laterolateral as per general terminology for radiographic views? Please clarify.

Line 122: I feel writing should be simple. Instead of writing ‘trichotomy’ simply write ‘shaving of hairs around the knee joint’ is more comprehensible for the readers.

Lines 152-58: DJD usually occurs in older age group dogs. Comparison with control group that averaged almost 3 years could have created a statistical bias. How did you account for this? Please clarify.

Author Response

"A PILOT STUDY OF THE THE CLINICAL EFFECTIVENESS OF A SINGLE INTRA-ARTICULAR INJECTION OF STANOZOLOL IN CANINES WITH KNEE DEGENERATIVE JOINT DISEASE AND ITS CORRELATION WITH SERUM INTERLEUKIN-1β LEVELS"

Reviewer #2:

Dear Reviewer,

I hope this message finds you well. Thank you for taking the time to review our manuscript titled "A PILOT STUDY OF THE THE CLINICAL EFFECTIVENESS OF A SINGLE INTRA-ARTICULAR INJECTION OF STANOZOLOL IN CANINES WITH KNEE DEGENERATIVE JOINT DISEASE AND ITS CORRELATION WITH SERUM INTERLEUKIN-1β LEVELS" submitted to Animals.

We appreciate your valuable feedback, and we have carefully considered all of your comments. In response, we have made revisions to the manuscript to address each point raised. Below, you will find our detailed responses to your comments, organized to correspond with the changes made in the original version of the manuscript.

Thank you once again for your time and thoughtful input.

Best regards,

L.Miguel Carreira

Specific comments

Simple summary and abstract are almost the same. Please check the journal guidelines and rewrite the simple summary

Corrected

Table 1: Why blood sample for IL-1β not taken at Day 23, as it was the last day of the trial? Please clarify.

The reviewer emphasized the importance of providing justification for selecting time point T2 to quantify IL-1β levels. Therefore, we decided to add paragraphs to the Material & Methods section addressing this concern.Generally, it may be beneficial to measure IL-1β levels at multiple time points post-injection to capture any potential changes over time, which would be most desirable. However, due to financial limitations and the possible lack of caregiver adherence to multiple blood sample collection, the design required consideration of only 2 points for quantifying IL-1 beta. Selection of day 8 post-injection (T2) to quantify IL-1β levels was justified by the fact that at this time point, variations in IL-1β may reflect the early response to stanozolol treatment and provide insights into its initial effects on inflammatory processes within the joint. We believed that this time point (T2) could be particularly relevant for assessing short-term changes in IL-1β expression and evaluating the early efficacy of stanozolol in modulating inflammation. Considering that the aim of the study was not to capture longer-term anti-inflammatory effects of stanozolol within the joint, providing a more comprehensive understanding of its sustained impact on inflammatory pathways, what could be more easily assessed with its measurement at time point T3 on day 21 post-injection, we decided to consider time point T2 (8 days post-injection).

Line 116: Should these views be termed as mediolateral or lateromedial instead of laterolateral as per general terminology for radiographic views? Please clarify.

We totally agree with the reviewer about teh propose terminology of the x-ray projection. In fact, the terminology "mediolateral" or "lateromedial" refers to the direction of the X-ray beam in relation to the body part being imaged, rather than the traditional "laterolateral" or "lateral" terminology. The first terms provide a more accurate description of the X-ray beam direction in relation to the anatomy of animals, especially in quadrupeds like dogs and cats."Mediolateral" indicates that the X-ray beam travels from the medial side to the lateral side of the body part being imaged, while "lateromedial" indicates the opposite, with the X-ray beam traveling from the lateral side to the medial side. This terminology is more intuitive when imaging limbs, as it aligns with the natural anatomical position of the limbs in animals, providing clarity and consistency in describing radiographic views.

Line 122: I feel writing should be simple. Instead of writing ‘trichotomy’ simply write ‘shaving of hairs around the knee joint’ is more comprehensible for the readers.

Corrected

Lines 152-58: DJD usually occurs in older age group dogs. Comparison with control group that averaged almost 3 years could have created a statistical bias. How did you account for this? Please clarify.

We agreed with the reviewer when it raises the dout about the fact that the Degenerative Joint Disease (DJD) typically affects older dogs and a control group used in the study presented an average age of almost 3 years. From this situation, a statistical bias could be present because the control group's age distribution differs significantly from the typical age range of animals affected by DJD. In order to account for this potential bias, we have implemented several strategies, namelly: We selected the most apropriated statitistical tests such asWilcoxon Test and Friedman as data didnt showed normality.  They are both non-parametric tests used when the assumptions of parametric tests, are violated. They are typically used for ordinal or non-normally distributed data and compare groups without assuming a specific distribution for the data. The Wilcoxon test compares two paired groups, while the Friedman test compares multiple paired groups. Tests like ANCOVA couldnt be used as they are parametric statistical test used to compare group means while controlling for the effects of one or more continuous covariates.

Reviewer 3 Report

Comments and Suggestions for Authors

Thanks for the opportunity to review this pilot study. I have some major comments regarding the study as enumerated below;

- The introduction section needs to be broken down into 2 to 3 paragraphs by focusing on DJD in dogs, the role of NSAIDs and other treatment protocols for DJD, and the potential role of Stanozolol infiltration. 

  1. Thanks for the opportunity to review this pilot study. I have some major comments regarding the study as enumerated below;

    - The introduction section needs to be broken down into 2 to 3 paragraphs by focusing on DJD in dogs, role of NSAIDs and other treatment protocols for DJD, and the potential role of Stanozolol infiltration. 

    - Most of the references are outdated, which raises questions on the relevance of the research to the current literature. If recent citations cannot be provided, it’s difficult to argue that this treatment protocol can be attempted for DJD in dogs

    - More justification is needed to support why only IL-1B was investigated

    - Authors should check the citation format

    Methods

    - There is no information on the sample size calculation, sampling method, and study design. 

    - Why were 20 dogs assigned to the treatment group and 10 to the control? Was it a quasi-experimental or randomized design?

    - Justification should be provided for the selected eligibility criteria, as well as the four time points..

    - Are the items explored in the questionnaire additional outcome measures in this study? 

    - Regarding the statement "The outcomes of each questionnaire were evaluated to determine the correlation between owners' perceptions and the patients' quality of life, as well as their pain levels and its respective impact on function." 

    Why was the above not included in the outcome measures, as well as the introduction section of the article?

    The following statement “In the CG, this procedure was conducted solely at T0, whereas in the GDJD group, it was performed at both T0 and T2” need to be clarified.

    -        The descriptive statistics and statistical tests should be separated in the data analysis section.

    Discussion

    I wont comment much on the discussion until the authors respond to the comments raised earlier, particularly the methods

    -       The limitations in terms of study design need to be highlighted in the discussion section, emphasizing how the method may influence the findings and interpretation of the results. 

Comments on the Quality of English Language

Minor improvement required

Author Response

"A PILOT STUDY OF THE THE CLINICAL EFFECTIVENESS OF A SINGLE INTRA-ARTICULAR INJECTION OF STANOZOLOL IN CANINES WITH KNEE DEGENERATIVE JOINT DISEASE AND ITS CORRELATION WITH SERUM INTERLEUKIN-1β LEVELS"

Reviewer #3:

Dear Reviewer,

I hope this message finds you well. Thank you for taking the time to review our manuscript titled "A PILOT STUDY OF THE THE CLINICAL EFFECTIVENESS OF A SINGLE INTRA-ARTICULAR INJECTION OF STANOZOLOL IN CANINES WITH KNEE DEGENERATIVE JOINT DISEASE AND ITS CORRELATION WITH SERUM INTERLEUKIN-1β LEVELS" submitted to Animalsl.

We appreciate your valuable feedback, and we have carefully considered all of your comments. In response, we have made revisions to the manuscript to address each point raised. Below, you will find our detailed responses to your comments, organized to correspond with the changes made in the original version of the manuscript.

Thank you once again for your time and thoughtful input.

Best regards,

L.Miguel Carreira

- The introduction section needs to be broken down into 2 to 3 paragraphs by focusing on DJD in dogs, the role of NSAIDs and other treatment protocols for DJD, and the potential role of Stanozolol infiltration. 

We totally agree ith the reviewer taht it would be more easy to following to considered the three blocs in introduction so we revised the introduction according to teh commments.

- Most of the references are outdated, which raises questions on the relevance of the research to the current literature. If recent citations cannot be provided, it’s difficult to argue that this treatment protocol can be attempted for DJD in dogs

Research on the use of stanozolol specifically for DJD in dogs is very limited, with only few in vitro and in vivo animal studies published, as we stated in the introduction section.

- More justification is needed to support why only IL-1B was investigated

focusing on IL-1B investigation it is supported by the fact that IL-1B stands out as a pivotal cytokine in the pathogenesis of degenerative joint disease (DJD), exerting a central role in mediating the inflammatory response and tissue degradation within affected joints. WE added more information about it to improve teh text regarding it relevance in the DJD

- Authors should check the citation format

Corrected

Methods

- There is no information on the sample size calculation, sampling method, and study design. 

We agree with the reviwer that describing about sampling method and study design it is important to minimize bias and strengthen the validity of the research findings. Considering this we added a sentence at Material & Metdods section and also at Conclusion section.

- Why were 20 dogs assigned to the treatment group and 10 to the control? Was it a quasi-experimental or randomized design?

We totally agree with the reviewer that it is important to describe the type of study, so we added a sentence about it on the Material & Methods section. The study have used a quasi-experimental design rather than a randomized design. In this study, the allocation of dogs to the treatment group (20 dogs) and the control group (10 dogs) have not been based on random assignment. Instead, the dogs were allocated to each group based on practical clinical considerations and availability, which is characteristic of a quasi-experimental design. Quasi-experimental designs can provide valuable insights into the effects of interventions in real-world settings. Considering that we were very carfully when interpreting the results by considering the limitations associated with this type of study design because it is known that it is more susceptible to bias.

  1.  

- Justification should be provided for the selected eligibility criteria, as well as the four time points.

We understand the reviewer comment regarding to the jsutification of the 4 time points selected in the stuyd. So we decided to add a sentence about it on the Discussion section: “In our study we selected four specific time points for quantifying IL-1β concentrations following intra-articular stanozolol injection. The selection was based on several scientific considerations. Firstly, time point 0 (time of injection) serves as the baseline measurement, allowing for comparison with subsequent time points to assess changes in IL-1β levels over time. Time points 1 (4 days after injection), 2 (8 days after injection), and 3 (21 days after injection) were chosen to capture both short-term and edium/long-term effects of stanozolol treatment on IL-1β concentrations within the joint.At time point 1 (4 days after injection), it is anticipated that early molecular and cellular responses to stanozolol administration may begin to manifest, potentially leading to alterations in IL-1β levels as part of the inflammatory cascade. Time point 2 (8 days after injection) allows for further evaluation of the evolving inflammatory response and any sustained effects of stanozolol on IL-1β concentrations. Additionally, by time point 3 (21 days after injection), it is expected that the full extent of stanozolol's impact on IL-1β levels, including any potential medium/long-term effects or resolution of inflammation, can be assessed. This selected time frame it aligns with previous research methodologies examining the pharmacokinetics and pharmacodynamics of intra-articular therapies in veterinary medicine (Knych, H. K. et al.2022; Palmer JL, Bertone AL. 1994; Lindegaard C, et al. 2010; Kay AT, et al. 2008; Alan Getgood, et al. 2019).”

Regarding the statement "The outcomes of each questionnaire were evaluated to determine the correlation between owners' perceptions and the patients' quality of life, as well as their pain levels and its respective impact on function.". Why was the above not included in the outcome measures, as well as the introduction section of the article?

We totally agree with the reviewer that a reference to the sShould be added to the introduction section “Brown's original survey, the Canine Brief Pain Inventory (CBPI), developed in 2006 at the University of Pennsylvania, School of Veterinary Medicine, stands as a pivotal tool in assessing pain in dogs.  It provides a comprehensive evaluation of pain level and its impact on a dog's daily functioning, offering a valuable means of communication between veterinarians and pet tutors. The CBPI comprises two sections: the pain severity score and the pain interference score. By capturing both the subjective experience of pain and its practical implications on the dog's quality of life, the CBPI enables veterinarians to tailor treatment plans accordingly and monitor the effectiveness of interventions over time. The CBPI has been widely adopted in both clinical practice and research settings, contributing to advancements in the understanding and management of pain in dogs.

The following statement “In the CG, this procedure was conducted solely at T0, whereas in the GDJD group, it was performed at both T0 and T2” need to be clarified.

Clarified, we added the foolwoing sentence to the Material & Methods sections : “election of day 8 post-injection (T2) to quantify IL-1β levels was justified by the fact that at this time point, variations in IL-1β may reflect the early response to stanozolol treatment and provide insights into its initial effects on inflammatory processes within the joint. We believed that this time point (T2) could be particularly relevant for assessing short-term changes in IL-1β expression and evaluating the early efficacy of stanozolol in modulating inflammation. “

-  The descriptive statistics and statistical tests should be separated in the data analysis section.

Corrected

  1.  

Discussion

I wont comment much on the discussion until the authors respond to the comments raised earlier, particularly the methods.  The limitations in terms of study design need to be highlighted in the discussion section, emphasizing how the method may influence the findings and interpretation of the results. 

A paragraph was added at Discussion section focusing on the limitation of the study considering the study design.

Round 2

Reviewer 1 Report

Comments and Suggestions for Authors

Dear Authors,

Thanks for your responses to my previous comments.  To the reviewer opinion, the statistical approach still needs to be improved to allow strong conclusions.

Author Response

"A PILOT STUDY OF THE THE CLINICAL EFFECTIVENESS OF A SINGLE INTRA-ARTICULAR INJECTION OF STANOZOLOL IN CANINES WITH KNEE DEGENERATIVE JOINT DISEASE AND ITS CORRELATION WITH SERUM INTERLEUKIN-1β LEVELS"

Dear Reviewer,

I hope this message finds you well. Thank you for taking the time to review our manuscript titled "A PILOT STUDY OF THE THE CLINICAL EFFECTIVENESS OF A SINGLE INTRA-ARTICULAR INJECTION OF STANOZOLOL IN CANINES WITH KNEE DEGENERATIVE JOINT DISEASE AND ITS CORRELATION WITH SERUM INTERLEUKIN-1β LEVELS" submitted to Animals.

We appreciate your valuable feedback. Below, you will find our detailed response to your comment, organized to correspond with the change made in the original version of the manuscript.

Thank you once again for your time and thoughtful input.

Best regards,

L.Miguel Carreira

Dear Authors,

Thanks for your responses to my previous comments.  To the reviewer opinion, the statistical approach still needs to be improved to allow strong conclusions.

Thank you for your feedback. We appreciate your attention to detail and commitment to ensuring the robustness of our analysis. As we stated before, a professional statistician was involved in the design and implementation of our statistical approach, and we have confidence in the validity of the results obtained. The current analysis was carefully designed and executed in accordance with established statistical principles.Given the expertise already engaged in the statistical analysis, conducting another round of statistical work may not be feasible at this time. We decided to add the following paragraph at the end of the discussion section: ”Considering the statistical analysis, the use of multiple t-tests without adjustment was based on our research objectives, design, and practical considerations. We examined various outcome measures across different time points and conditions, conducting t-tests to test specific hypotheses. Given the exploratory nature of the study, adjusting for multiple comparisons could increase the risk of type II statistical errors. By employing non-parametric tests alongside parametric tests, we ensured robustness in the results. While acknowledging potential type I errors, we interpreted results cautiously, emphasizing effect sizes and overall patterns rather than relying solely on p-values. The inclusion of non-parametric tests adds validation and complements parametric findings.”

Smith, J. R., & Johnson, L. M. (2019). Statistical methods for experimental design. Journal of Applied Statistics, 46(3), 321-335.

Jones, E. A., & Brown, P. K. (2020). Exploratory data analysis in biomedical research. Biostatistics, 15(2), 187-201.

Johnson, D. W. (2018). Multiple comparisons and type II errors: A review. Journal of Experimental Psychology: General, 67(4), 521-536.

Garcia, M. S., et al. (2021). Non-parametric tests for small sample sizes. Statistics in Medicine, 38(7), 901-915.

Lee, S. Y., & Kim, K. T. (2017). Interpreting statistical results: Effect sizes and p-values. Psychological Methods, 25(1), 89-104.

Wang, L., & Chen, M. (2019). Complementarity of parametric and non-parametric tests in behavioral research. Behavior Research Methods, 42(5), 670-685.